# Using Machine Learning to Develop and Validate an In-Hospital Mortality Prediction Model for Patients with Suspected Sepsis

**DOI:** 10.3390/biomedicines10040802

**Published:** 2022-03-29

**Authors:** Hsiao-Yun Chao, Chin-Chieh Wu, Avichandra Singh, Andrew Shedd, Jon Wolfshohl, Eric H. Chou, Yhu-Chering Huang, Kuan-Fu Chen

**Affiliations:** 1Department of Emergency Medicine, Linkou Chang Gung Memorial Hospital, No. 5, Fu-Shin Street, Gueishan Village, Taoyuan 333423, Taiwan; bravojim@adm.cgmh.org.tw; 2Clinical Informatics and Medical Statistics Research Center, Chang Gung University, Taoyuan 33302, Taiwan; wujinja@mail.cgu.edu.tw; 3Department of Emergency Medicine, Keelung Chang Gung Memorial Hospital, Keelung 20401, Taiwan; smile3196@cgmh.org.tw; 4Department of Emergency Medicine, Baylor Scott and White All Saints Medical Center, Fort Worth, TX 76104, USA; ashedd@ies.healthcare (A.S.); jon.wolfshohl1@bswhealth.org (J.W.); eric.chou@bswhealth.org (E.H.C.); 5Department of Emergency Medicine, Baylor University Medical Center, Dallas, TX 76104, USA; 6Division of Pediatric Infectious Diseases, Linkou Chang Gung Memorial Hospital, No. 5, Fu-Shin Street, Gueishan Village, Taoyuan 333423, Taiwan; ychuang@cgmh.org.tw; 7Community Medicine Research Center, Keelung Chang Gung Memorial Hospital, Keelung 20401, Taiwan

**Keywords:** biomarker, logistic regression, machine learning, mortality prediction, sepsis

## Abstract

Background: Early recognition of sepsis and the prediction of mortality in patients with infection are important. This multi-center, ED-based study aimed to develop and validate a 28-day mortality prediction model for patients with infection using various machine learning (ML) algorithms. Methods: Patients with acute infection requiring intravenous antibiotic treatment during the first 24 h of admission were prospectively recruited. Patient demographics, comorbidities, clinical signs and symptoms, laboratory test data, selected sepsis-related novel biomarkers, and 28-day mortality were collected and divided into training (70%) and testing (30%) datasets. Logistic regression and seven ML algorithms were used to develop the prediction models. The area under the receiver operating characteristic curve (AUROC) was used to compare different models. Results: A total of 555 patients were recruited with a full panel of biomarker tests. Among them, 18% fulfilled Sepsis-3 criteria, with a 28-day mortality rate of 8%. The wrapper algorithm selected 30 features, including disease severity scores, biochemical parameters, and conventional and few sepsis-related biomarkers. Random forest outperformed other ML models (AUROC: 0.96; 95% confidence interval: 0.93–0.98) and SOFA and early warning scores (AUROC: 0.64–0.84) in the prediction of 28-day mortality in patients with infection. Additionally, random forest remained the best-performing model, with an AUROC of 0.95 (95% CI: 0.91–0.98, *p* = 0.725) after removing five sepsis-related novel biomarkers. Conclusions: Our results demonstrated that ML models provide a more accurate prediction of 28-day mortality with an enhanced ability in dealing with multi-dimensional data than the logistic regression model.

## 1. Introduction

Sepsis was first defined as a documented or suspected infection with systemic inflammatory response syndrome (SIRS) [1]. However, this conventional definition was abandoned in 2016, and “sepsis-3” serves as a new definition of “sepsis”, intending to increase prognostic accuracy [2]. The incidence of sepsis has been steadily increasing since the early 2000s [3] and has been estimated at approximately 6 cases per 100 adult hospitalizations per year, which has not changed significantly in the last decade in both the United States and the world [4,5]. The sepsis fatality rate was approximately 30% in the early 2000s [6] with a decreasing trend in mortality between 2009 and 2011, but not after 2011 [7].

Predicting the outcomes of patients with infection continues to be a topic of interest. However, no single biomarker can be used to predict sepsis satisfactorily [8]. Efforts to combine predictors, such as the sequential organ failure assessment (SOFA) score [9] and the chills, hypothermia, anemia, red cell distribution width and malignancy (CHARM) score [10] are evidence that researchers hope to develop a feasible and accurate prediction model for clinical utility. However, their capacity to predict mortality in patients with infection is inadequate (area under the receiver operating characteristic curve (AUROC) of SOFA ranged 0.66–0.84) [11,12,13,14].

In the past few decades, regression models have dominated the field of clinical prediction modelling, largely because of their simplicity in application and easy interpretation. However, regression models do not perform well in nonlinear distributions, complicated interactions, and high dimensionality, and the results may be invalid when the expectations are not met. Some interesting activities stemming from the fields of machine learning (ML) have varying success in the prediction of mortality caused by sepsis (Appendix A) [15,16]. A meta-analysis suggested that ML approaches performed better than the existing scoring systems, such as SIRS, National Early Warning Score (NEWS), Modified Early Warning Score (MEWS), qSOFA, and SOFA in predicting the onset of sepsis [17]. We aimed to develop and validate a machine learning-based model to predict 28-day mortality for patients with infection in a prospective, multi-center, hospital-based cohort study, which included demographics, comorbidities, clinical signs and symptoms, disease severity, hemodynamic indicators, biochemical profiles, and relevant biomarker panels.

## 2. Materials and Methods

### 2.1. Study Design and Study Population

This prospective observational study was conducted in the emergency departments (EDs) of three secondary and tertiary teaching hospitals in Northern Taiwan, from February 2014 to September 2017. The study was approved by the Institutional Review Board of the Chang Gung Memorial Hospital (approval number: 201800543B0). Patients who met the selection criteria were enrolled after signing an informed consent form. Patients with acute infection who required hospital admission for intravenous antibiotics treatment were eligible in the study, while patients who were transferred from other hospitals or hospitalized within the previous 2 weeks due to infectious disease, had received antibiotics of unknown classes and dosing schedules, received a blood transfusion within 24 h, or received renal dialysis within 12 h were excluded. Discordant diagnoses were resolved by an expert panel to confirm their eligibility. We followed a prospective-specimen-collection, retrospective-blinded-evaluation (PRoBE) design [18] in which biologic specimens are collected prospectively from the target population.

### 2.2. Measurements

Blood samples were tested for complete cell counts, biochemical parameters, coagulation function, liver and renal function, immunological function, and sepsis-related novel biomarkers, such as acute-phase proteins: Pentraxin-3; cytokines and chemokines: tumor necrosis factor-alpha (TNF-α), interleukin-6 (IL-6), interleukin-8 (IL-8), interleukin-10 (IL-10), and interferon-gamma (IFN-γ); cell surface markers: cluster of differentiation-14 (sCD14), cluster of differentiation-64 (sCD64) and the soluble cluster of differentiation-163 (sCD163); receptor markers: triggering receptor expressed on myeloid cells-1 (TREM-1); and endothelial damage markers: angiopoietin-2, E-selectin, P-selectin, intercellular adhesion molecule-1 (ICAM-1), and vascular cell adhesion protein-1 (VCAM-1). The sepsis-related novel biomarker tests were performed using the multiplex platform of Bio-Plex ProTM Assays (Bio-Rad Laboratories, Hercules, CA, USA). For specimens with insufficient volume for all biomarker examinations, novel sepsis-related biomarkers were prioritized for testing. Various disease scoring systems, including SOFA and ΔSOFA scores (change in total SOFA score between the ED visit and the baseline value; Supplemental boar), CHARM score, NEWS, MEWS, and SIRS were calculated to assess the severity of illness. A detailed history of present illness, vital signs, basic biochemical profile, chief complaints, comorbidities, and disease severity during hospitalization were obtained by dedicated research personnel. Sepsis-1 was defined as infection that fit two or more criteria of SIRS. Sepsis-3 and septic shock were defined according to the third international consensus for the definition of sepsis and septic shock (Appendix A) [19]. The clinical gestalt was obtained prospectively by inquiring of the primary care physician in the EDs about the estimated possibility of death on a five-point Likert scale. The primary outcome was defined as the 28 d mortality, which was obtained through electronic medical records for in-hospital patients and telephone follow-up survey for those discharged patients.

### 2.3. Data Partitioning

We adopted a stratified random sampling strategy to divide our dataset into a 70% training dataset and 30% testing dataset stratified by outcome (28-day mortality). Owing to the relatively small sample size, we used a three-fold cross-validation method to fine-tune our prediction models on the training dataset. Briefly, we randomly divided the training dataset into three folds of equal size. For each round of cross-validation, two of these folds were used to train our models, and the third was used to evaluate the performance and fine-tune the hyperparameters.

### 2.4. Feature Engineering and Feature Selection

Based on the characteristics of the datasets, two types of feature engineering methods were applied in this study: multiplication of features to amplify the effect and centering of data with U-shaped relationships with 28-day mortality via visual identification of the turning points in locally weighted scatter-plot smoothing curves. The centered features were subsequently squared to amplify the signals in the models. For feature selection, we adopted one of the wrapper algorithms, the “Boruta” algorithm, to rank the features associated with mortality in the random forest (RF) models [20].

### 2.5. Data Preprocessing

Features with missing values were evaluated for the missing mechanism, followed by their association with mortality. The missing values were then replaced with medians for continuous features and modes for categorical features [21]. We normalized the continuous features to facilitate model training.

### 2.6. Machine Learning-Based Models

In this study, we evaluated seven ML algorithms, including four tree and ensemble-based models: extreme gradient boosting, conditional random forest, random forest, and random forest generator; one distance-based model, support vector machine; and two neural network-based models: artificial neural networks and deep neural networks, and compared them with one regression-based model: logistic regression (Appendix A).

### 2.7. Performance Evaluation and Model Interpretation

The AUROC was used to assess the performance of the constructed models [22]. We also used the Shapley additive explanation value to interpret the output of the models that helped us to understand the direction and strength of a single feature in the final model [23,24].

### 2.8. Sensitivity Analysis

To increase the practicality of the prediction model, we performed sensitivity analyses by (1) removing the sepsis-related novel biomarkers from the candidate features to make the model more practical and (2) using only the top five candidate features in the model. We also evaluated whether the synthetic minority oversampling technique (SMOTE) could improve the performance of our models by up- or down-sampling methods to handle the potential data imbalance on the training dataset [25].

### 2.9. Statistical Analysis

Data on demographics and clinical characteristics are presented as mean (standard deviation) or median (interquartile range) for continuous features and counts and percentages for discrete features. Descriptive statistics were compared using the Wilcoxon rank-sum test for continuous data comparisons and the chi-square test for categorical data. All statistical tests were two-sided, and statistical significance was set at *p*-value < 0.05. Statistical and ML algorithms were performed using R (version 4.0.1, Vienna, Austria) with the caret package. Comparisons of the AUROC between prediction models were made using the DeLong method of the pROC package [22]. To achieve a 0.8 power with 0.95 accuracy, the minimal number of test groups needs to be 100 in supervised learning methods. We reported our study according to the Transparent Reporting of a multivariable prediction model for Individual Prognosis or Diagnosis Statement (TRIPODS) [26].

## 3. Results

### 3.1. Study Population and Characteristics

During the study period, 555 patients with a full panel of biomarker tests were recruited. Patients were predominantly males with a mean age of 62.48 ± 17.55 years, in which 75% fulfilled the Sepsis-1 criteria and 18% fulfilled the Sepsis-3 criteria. Vital signs, disease severity scores, and most of the sepsis-related novel biomarkers and indices demonstrated trends in accordance with disease severity (Table 1 and Appendix A).

The 28-day mortality rate of the study cohort was 8%. Non-survivors tended to be older and male, with a higher proportion of having malignancies, presenting with a lower GCS of less than 15, increased respiratory rate and poorer oxygen saturation, abnormal hematological profiles, such as lower hematocrit and higher red cell distribution width, and were more likely to have sepsis-3 and septic shock (Table 2). The mean levels of the sepsis-related novel biomarkers measured at baseline were higher in non-survivors than in survivors (Appendix A). Patients with missing data had no significant difference in 28-day mortality compared to those without missing data.

### 3.2. Machine Learning Development and Evaluation

First, we divided the overall dataset into training and testing sets with sample sizes of 389 and 166, respectively, which contained 219 features (Appendix A). Among them, 70 features were continuous, and 12 features had U-shaped distributions and were centered before feature selection, including pulse rate, respiratory rate, systolic blood pressure, diastolic blood pressure, mean arterial pressure, blood sugar, activated partial thromboplastin time, phosphorous, calcium, protein C, bicarbonate, and white blood cell count (Appendix A).

After centering for the U-shaped association, 30 features were selected using the wrapper algorithm around the random forest (RF) models, and their mean weighted contribution and AUROC for prediction are listed in Appendix A. Among these selected features, four were disease severity scores (SOFA, SOFA-respiratory, SOFA-coagulation, and ΔSOFA), nine were arterial blood gas parameters (AaDO_2_, pH, total CO_2_, ABE, SBC, SBE, HCO_3_, pCO_2_, and FiO_2_), five were sepsis-related novel biomarkers (IL-8, IL-6, angiopoietin-2, E-selectin, and VCAM1), and the rest were routine biochemical and conventional biomarkers (D-dimer, fibrin degradation products, procalcitonin, and lactate). Among these candidate features, SOFA score, normalized IL-8, D-dimer, cortisol, and albumin had the highest univariable AUROC of 0.74–0.83. In addition, although not selected by the wrapper algorithm, the clinician’s gestalt also had an acceptable performance in predicting 28-day mortality (AUROC: 0.83; 95% confidence interval (CI): 0.69–0.90).

The top 30 features were then used to determine the best model (Appendix A). In the testing dataset, RF performed best, with an AUROC of 0.96 (95% CI: 0.93–0.98) when all 30 selected features were used (number of trees: 500, and number of variables randomly sampled as candidates at each split: 34). After removing five sepsis-related novel biomarkers, RF remained the best-performing model, resulting in a similar AUROC of 0.95 (95% CI: 0.91–0.98, *p* = 0.725). By further reducing the number of features, the prediction performance began to decline. eXtreme Gradient Boosting performed best with an AUROC of 0.86 if only the top five features were used for model input (95% CI: 0.76–0.97) (Table 3).

### 3.3. Model Interpretation

Figure 1 shows the importance of the independent features ranked in a descending order in the final Random Forest model. The top five most important features in predicting 28-day mortality are the SOFA score, IL-8, D-dimer, IL-6, and angiopoietin-2. In general, the higher the levels of these biomarkers and disease scores, the more positive the impacts were in predicting 28-day mortality. On the contrary, the lower the albumin and platelet levels observed, the greater the risk of 28-day mortality. These observations were compatible with the simple univariate analysis (Appendix A).

### 3.4. Comparison between the Best Machine Learning and Traditional Scoring Systems

Compared with all the traditional scoring systems, RF performed best in predicting 28-day mortality on the testing dataset (AUROC: 0.96; 95% CI: 0.93–0.98, *p* < 0.001; Figure 2), and the AUROC remained high after removing the five sepsis-related biomarkers (AUROC: 0.95; 95% CI: 0.91–0.99). Among the seven traditional scoring systems, the CHARM score demonstrated the second-best performance with an AUROC of 0.86 (95% CI: 0.79–0.91) for 28-day mortality prediction, whereas SIRS performed the worst (AUROC: 0.53; 95% CI: 0.40–0.77).

### 3.5. Imbalance Data Management

By applying SMOTE for both the up-sampling and down-sampling procedures, we found that the process did not improve the performance. Therefore, imbalanced processing was not adopted in this study (Appendix A).

## 4. Discussion

In this prospective hospital-based cohort study, ensemble-based ML models, especially the random forest (RF) model, outperformed deep learning and logistic regression models and other traditional scoring systems in the prediction of 28-day mortality for patients with infection. We demonstrated that the ML models could be developed incorporating conventional features to assist the daily practice in the frontline health care settings. With 25 conventional features, the RF model had an AUROC of up to 0.95 in predicting 28-day mortality on the testing dataset. Many single biomarkers, such as IL-8, albumin, and D-dimer, were also found to have predictive power similar to that of the SOFA score. Sepsis-related novel biomarkers, including IL-8, IL-6, and angiopoietin-2, were included in the final models, but could also be substituted by other features without significant impact on the performance.

In our study, we demonstrated that RF can be used to rank the importance of features and derive a powerful prediction model with complicated interactions between features. RF can generate hundreds of decision trees to fit the dataset. By averaging the variances in the number of trees, RF reduces the high variance derived from a single tree. RF enables the evaluation of more features and interactions compared to traditional modelling approaches. A similar study that predicted 28-day mortality of ED patients with sepsis using real-world data carried out by Taylor et al. also demonstrated that the RF model performed better than the logistic regression model (AUROC of 0.86 vs. 0.76) [15].

The SOFA score was an important feature in our feature selection process and ML modelling in predicting 28-day mortality for patients with infection, in accordance with many other retrospective studies (pooled AUROC of 0.75–0.78) [12,19,27,28]. Furthermore, the SOFA respiratory score was selected in addition to the total SOFA score in our final RF model. Our findings suggest that respiratory dysfunction is an important predictor of mortality in patients with infection, which is supported by many other studies [5,29]. In addition, the qSOFA score, which was developed to screen for patients with possible sepsis-3, also contains the respiratory rate [19]. We believe that respiratory dysfunction contributes more to sepsis-associated mortality and should be considered an important risk factor in future research.

Several commonly measured biomarkers in clinical settings were selected for our final model. D-dimer is a fibrin degradation protein fragment that is formed after a blood clot is degraded by fibrinolysis. Severe infection may lead to the activation of an inflammatory cascade that can trigger this coagulopathy process [30]. Furthermore, the low serum albumin level in the acute phase of sepsis may be due to the inhibition of the albumin gene caused by TNF-α overexpression during inflammation [31]. This decrease in albumin level was significantly associated with the risk of death in septic patients [32]. Additionally, the presence of a large amount of lipopolysaccharides during sepsis activates the hypothalamus–pituitary–adrenal axis and the sympathoadrenal system, which leads to an increased output of cortisol and catecholamines, and ultimately an elevated serum lactic acid level, which has been widely recognized as a marker of tissue hypoxia or hypoperfusion and increased risk of multiple organ dysfunction syndrome [33].

Inflammation-related biomarkers, such as IL-8 and IL-6, were predictive of 28-day mortality both in the univariate analysis and in the final RF model. As a single biomarker, IL-8 has a similarly acceptable performance as the SOFA score (AUROC 0.83 vs. 0.82) and has been associated with 28-day mortality in a smaller study [34]. In contrast, the contributions of the other sepsis-related novel biomarkers were less prominent in our study. One possible reason is that the timing of sampling affected their predictive power, as many markers may only elevate at certain phases of disease progression and rapidly subside [35]. However, patients at different stages of infection severity were enrolled in this study. Therefore, we hypothesize that multiple measurements may capture the dynamic patterns of these indicators and better correlate with patient outcomes.

In this study, the clinician’s gestalt demonstrated moderate predictive accuracy and precision (AUROC: 0.83; 95% CI: 0.69–0.90). However, it was not selected in the RF models of our study. The wrapper algorithm ranks features that are selected in a model with high-order feature interactions, which indicates that the importance of clinical gestalt can be replaced by other features. Since the development of ML algorithms, researchers have been trying to enhance human experience and judgement by translating data into a language that the machine can understand [36]. In reality, clinician’s gestalt is seldom assessed together with the performance of machine-aided decision making, but if ever performed simultaneously and reported, the decision aid rarely outperformed human judgment [37].

Compared to previous studies that used retrospective electronic medical record data for model inputs, our study has the advantage of having good-quality data prospectively collected from three different emergency departments. This study still has some limitations. First, from the standpoint of clinical practicality, not all the information used for modelling inputs in this study is routinely collected from the emergency department. Nonetheless, the reduced model without those sepsis-related novel biomarkers still performs relatively well in our study. However, it is worth recognizing that the predictive capability of traditional score systems can be improved by applying ML algorithms to handle multidimensional features. Additionally, if laboratory and vital sign data can be collected repeatedly over time, this may allow more precise analysis to reflect the time-dependent nature of sepsis progression [38]. Lastly, the outcome that we adopted in this study, the 28-day mortality, might not be the best endpoint to improve the care of septic patients. Future models are needed to be developed to predict the response, such as fluid and inotropic agents for patients with suspected sepsis.

## 5. Conclusions

We derived and tested a multi-feature prediction model that estimates the mortality probability in adult patients with infection. We demonstrated that the ML model could enhance the ability to deal with multidimensional data and achieve excellent performance in outcome prediction. Our ML model does an excellent job in predicting mortality but at the expense of gathering a large number of data, which might not be cost-effective in real-world clinical settings. While it adds more understanding on enhancing the utility of health data through ML algorithms, critical issues remain to be resolved; further studies that evaluate the clinical utility of the developed models and predictors are required. Additionally, external validations to confirm whether these algorithms provide improved predictive value in identifying at-risk patients in various clinical settings is warranted before it can move forward to clinical application.

## Figures and Tables

**Figure 1 biomedicines-10-00802-f001:**
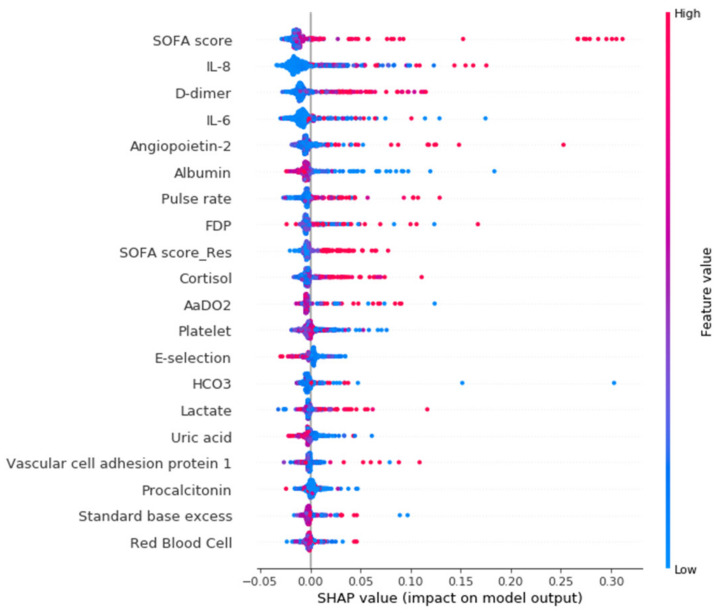
The Shapley Additive exPlanations (SHAP) summary plot of the final Random Forest models. The horizontal location of this SHAP plot demonstrates whether the effect of the value of that feature is associated with a higher or lower prediction of the model output, and the color indicates whether that feature is high (red) or low (blue) for that observation. SOFA, Sequential Organ Failure Assessment; SOFA score-Res, SOFA-respiratory; SOFA score-Coag, SOFA-coagulation; FDP, fibrin degradation products.

**Figure 2 biomedicines-10-00802-f002:**
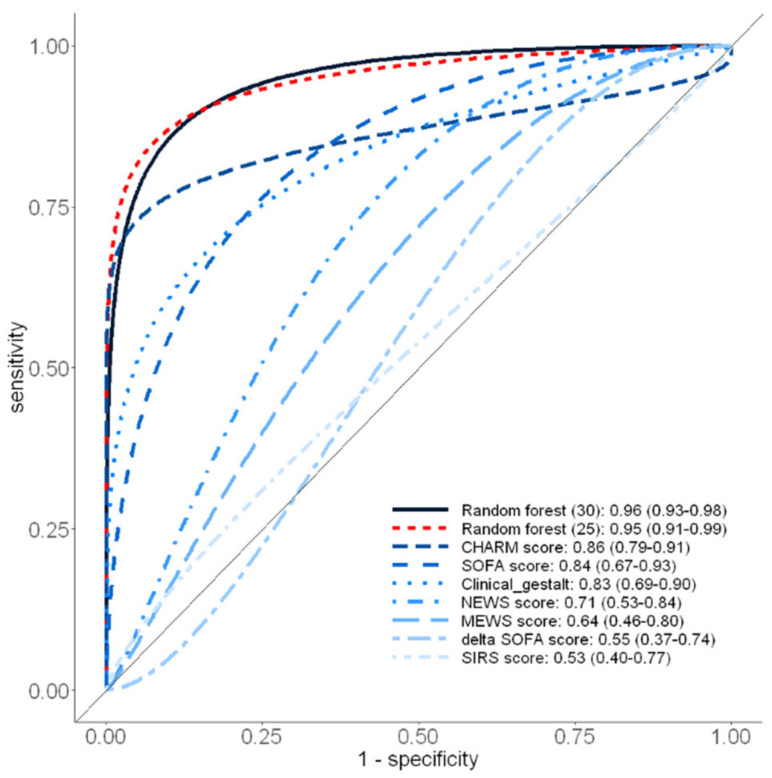
Area under the receiver operating characteristic curves derived from Random Forest, CHARM, SOFA, Clinical Gestalt, NEWS, MEWS, SIRS and ΔSOFA scores for the prediction of 28-day mortality on the testing dataset (*n* = 166). CHARM (Chills, Hypothermia, Anemia, Red Cell Distribution Width and Malignancy score); SOFA score (Sequential Organ Failure Assessment score); clinical gestalt (primary care physician’s estimation of the possibility of death); NEWS (National Early Warning Score); MEWS (Modified Early Warning Score); SIRS (Systemic Inflammatory Response Syndrome); delta SOFA score (change in total SOFA score between ED visit and the baseline value).

**Table 1 biomedicines-10-00802-t001:** Patient baseline demographics, comorbidities, and disease severity, stratified by stage of sepsis.

Features	Total	Sepsis-1	Sepsis-3	Septic Shock ^1^	Septic Shock ^2^
Numbers (%)/(Mean (SD)	555 (100)	418 (75.32)	101 (18.20)	58 (10.45)	7 (1.26)
Demographics					
Age (years)	62.48 (17.55)	63.17 (17.11)	68.58 (15.22)	69.17 (15.53	57.14 (20.38)
Male gender	350 (63.1)	271 (64.8)	69 (68.3)	44 (75.9)	6 (85.7)
Vital signs					
Body temperature (°C)	38.0 (1.26)	38.2 (1.25)	38.2 (1.26)	38.3 (1.24)	37.7 (0.68)
Pulse (bpm)	109 (21.42)	115 (19.82)	114 (22.88)	115 (22.35)	112 (23.23)
Respiratory rate (breaths/min)	21 (3.45)	21 (3.62)	22 (3.75)	22 (4.07)	22 (4.6)
SBP (mmHg)	137 (30.33)	138 (30.09)	128 (32.93)	125 (31.63)	100 (26.25)
DBP (mmHg)	78 (17.34)	78 (17.68)	72 (16.89)	71 (17.58)	62 (19.69)
GCS coma scale	15 (15–15)	15 (15–15)	15 (11–15)	14 (10.25–15)	15 (14.5–15)
Comorbidities					
Diabetes	220 (39.6)	168 (40.2)	42 (41.6)	24 (41.4)	0 (0.0)
Tumor	96 (17.3)	74 (17.7)	13 (12.9)	11 (19.0)	2 (28.6)
Chronic obstructive pulmonary disease	99 (17.8)	80 (19.1)	22 (21.8)	9 (15.5)	0 (0.0)
Congestive Heart Failure	42 (7.6)	32 (7.7)	8 (7.9)	2 (3.4)	0 (0.0)
Chronic Kidney Disease	45 (8.1)	32 (7.7)	16 (15.8)	9 (15.5)	0 (0.0)
Hemiplegia or paraplegia	80 (14.4)	60 (14.4)	18 (17.8)	10 (17.2)	1 (14.3)
Liver disease	83 (15.0)	64 (15.3)	12 (11.9)	7 (12.1)	0 (0.0)
Malignancy	155 (27.9)	117 (28.0)	30 (29.7)	21 (36.2)	3 (42.9)
Mild Liver Disease	70 (12.6)	52 (12.4)	10 (9.9)	6 (10.3)	0 (0.00
Cirrhosis Liver Disease	52 (9.4)	39 (9.3)	5 (5.0)	1 (1.7)	0 (0.0)
Site of infection					
Respiratory	221 (39.8)	169 (40.4)	42 (41.6)	21 (36.2)	4 (57.1)
Genitourinary	179 (32.3)	131 (31.3)	25 (24.8)	14 (24.1)	3 (42.9)
Skin	50 (9.0)	38 (9.1)	10 (9.9)	3 (5.2)	0 (0.0)
Abdominal	41 (7.4)	32 (7.7)	9 (8.9)	5 (8.6)	1 (14.3)
Central Nervous System	5 (0.9)	4 (1.0)	0 (0.0)	0 (0.0)	0 (0.0)
Unspecified	277 (49.9)	221 (52.9)	62 (61.4)	38 (65.5	6 (85.7)
Disease severity score (Median (IQR))					
SOFA	2 (1–4)	2 (1–4)	4 (3–6)	5 (3–7)	11 (7.5–13)
ΔSOFA	0 (−2–1)	0 (−2–1)	3 (2–5)	4 (3–6)	10 (7.5–12)
MEDS	6 (3–9)	6 (3–9)	8 (6–11)	8.5 (6–11)	9 (7–9)
CHARM	2 (1–3)	2 (1–3)	2 (1–3)	2 (1.25–3)	2 (1.5–2.5)
NEWS	6 (4–9)	7 (5–9)	8 (6–10)	8 (5.25–10)	8 (6–10.5)
MEWS	4 (2–5)	4 (3–6)	5 (3–6)	5 (3–6.75)	5 (3.5–5)
Outcomes (Number (%))					
ICU admission	27 (4.9)	23 (5.5)	6 (5.9)	6 (10.3)	3 (42.9)
In-hospital death	45 (8.1)	42 (10.04)	16 (15.84)	12 (20.69)	4 (57.14)

Sepsis-1 was defined as two or more criteria (score ≥ 2) of SIRS (Systemic Inflammatory Response Syndrome) plus suspected or documented infection. Sepsis-3 was defined as evidence of infection plus an acute increase of SOFA Score (ΔSOFA ≥ 2) compared to the baseline values. Two definitions of septic shock were applied: ^1^ (ΔSOFA ≥ 2 + Lactate > 18 mg/dL) and ^2^ (ΔSOFA ≥ 2 + Lactate > 18 mg/dL) + Vasopressor usage. GCS: Glasgow Coma Scale; SOFA: Sequential Organ Failure Assessment Score; MEDS: Mortality in Emergency Department Sepsis score; CHARM: Chills, Hypothermia, Anemia, Red Cell Distribution Width and Malignancy score; NEWS: National Early Warning Score; MEWS: Modified Early Warning Score.

**Table 2 biomedicines-10-00802-t002:** Patient characteristics, stratified by 28-day in-hospital mortality.

	28-Day In-Hospital Mortality	
Features Mean (SD)/*n* (%)	Survivor (*n* = 510)	Death (*n* = 45)
Demographics		
Age (years) *	61.87 (17.71)	68.88 (14.5)
Male	314 (61.9)	36 (75.0)
Underlying disease		
Malignancy *	127 (25)	28 (58.3)
Vital signs		
Body temperature (°C) *	38.02 (1.25)	37.46 (1.17)
SBP (mmHg)	137.59 (29.74)	131.27 (35.87)
DBP (mmHg)	77.96 (16.71)	75.81 (23.13)
Respiratory rate (breaths/min) *	20.51 (3.10)	23.96 (5.06)
Pulse (bpm) *	108.9 (20.19)	114.06 (30.86)
SPaO_2_ (%) *	93.6 (4.43)	90.11 (7.08)
GCS *	15 (15–15)	15 (10.75–15)
Hemogram and biochemical profile		
Hemoglobin (g/dL) *	12.21 (2.15)	10.64 (1.93)
Red Blood Cell (10^6^ μL) *^,₸^	4.17 (0.75)	3.74 (2.12)
RDW (%) *^,₸^	14.41 (2.02)	15.78 (2.05)
Band (% of WBC) *	1.7 (4.25)	5.49 (7.07)
Platelet (10^3^ μL) *^,₸^	213.39 (96.82)	153.77 (96.50)
AST (U/L) ^₸₸^	47.57 (114.29)	79.76 (106.31)
BUN (mg/dL) *^,₸₸^	20.95 (18.30)	38.55 (36.54)
Albumin (g/dL) *^,₸₸^	3.48 (0.54)	2.77 (0.68)
Uric acid (mg/dL) *^,₸₸₸^	5.48 (2.30)	6.85 (3.03)
Potassium (mEq/L) *	3.82 (0.59)	4.15 (0.84)
Phosphorous (mg/dL) *	2.92 (1.50)	3.9 (1.57)
Protein C ^₸₸₸^	726.69 (1048.14)	970.28 (1182.41)
Coagulation profiles		
Prothrombin Time (s) *^,₸₸^	13.94 (2.97)	16.09 (4.69)
INR *^,₸₸₸^	1.23 (0.27)	1.39 (0.35)
FDP (μg/mL) *^,₸₸₸^	18.94 (15.03)	34.84 (24.90)
Cortisol (μg/dL) *^,₸₸^	22.89 (16.56)	42.41 (33.66)
Gas profile		
AaDO_2_ (mmHg) *^,δ^	55.50 (30.36)	115.90 (163.33)
pH *^,δ^	7.41 (0.05)	7.33 (0.16)
Total CO_2_ (mmol/L) ^δ^	25.15 (4.19)	24.30 (8.05)
ABE (mmol/L) *^,δ^	−0.39 (3.59)	−2.80 (8.85)
SBC (mmol/L) *^,δδ^	23.34 (3.50)	19.84 (9.12)
SBE (mmol/L) *^,δ^	−0.58 (4.13)	−2.86 (9.61)
HCO_3_ (mmol/L) ^δ^	23.80 (3.89)	22.8 (8.04)
pCO_2_ (mmHg) *^,δ^	38.48 (8.39)	43.08 (14.14)
FiO_2_ *^,₸^	26.20 (11.21)	34.73 (25.73)
Conventional biomarkers		
Procalcitonin (ng/mL) *^,₸^	0.55 (0.09–5.18)	1.99 (0.36–26.2)
Lactate (mg/dL) *^,₸₸^	14.3 (10.47–20.92)	22.8 (15.45–37.9)
C reactive protein *^,₸₸^	80.04 (37–158.3)	120.86 (70.84–200.27)
D-dimer (ng/mL) *^,₸₸₸^	1183.5 (509.5–2374.25)	3609.5 (1592.5–10,000)
Disease severity		
Sepsis-3 *	87 (17.06)	14 (31.11)
Septic Shock *	48 (9.41)	10 (22.22)
CHARM *	2 (1–2)	3 (2–4)
Length of hospital stay (days) Median (IQR)	10 (7–16)	15.5 (6.25–24.75)
Clincial gestalt *	2.25 (0.71)	3.05 (0.87)

Missing data: ^₸^ <1%; ^₸₸^ 1~5%; ^₸₸₸^ 5~10%; ^δ^ 30~80%; ^δδ^ >80%; * *p*-value < 0.05. ABE: actual base excess; SBE: standard base excess; SBC: standard bicarbonate measurement: Septic shock defined as Sepsis-3 and lactate > 18 mg/dL.

**Table 3 biomedicines-10-00802-t003:** The area under the receiver operating characteristic curves of seven machine learning models when various features were selected.

Models	30 Selected Features		^δ^ 25 Selected Features (Remove Biomarkers)		^δδ^ Top 5 Features Only	
	Training	Testing	Training	Testing	Training	Testing
eXtreme Gradient Boosting	0.989 (0.981–0.997)	0.934 (0.887–0.980)	0.975 (0.960–0.990)	0.924 (0.876–0.972)	0.920 (0.880–0.960)	0.860 (0.755–0.965)
Conditional random forest	0.943 (0.916–0.969)	0.933 (0.889–0.977)	0.939 (0.910–0.968)	0.931 (0.890–0.972)	0.933 (0.898–0.967)	0.843 (0.721–0.965)
Random Forest	1.000 (1.000–1.000)	0.959 (0.927–0.983)	1.000 (1.000–1.000)	0.948 (0.913–0.977)	1.000 (1.000–1.000)	0.831 (0.724–0.924)
RANdom forest GEneRator	0.991 (0.984–0.999)	0.940 (0.899–0.981)	0.990 (0.982–0.998)	0.938 (0.896–0.980)	0.958 (0.933–0.983)	0.843 (0.734–0.952)
Support vector machine	0.977 (0.953–0.999)	0.881 (0.796–0.966)	0.921 (0.879–0.962)	0.871 (0.783–0.959)	0.999 (0.999–1.000	0.693 (0.491–0.895)
Neural network	0.894 (0.848–0.940)	0.821 (0.715–0.926)	0.878 (0.824–0.931)	0.713 (0.525–0.901)	0.894 (0.840–0.947)	0.800 (0.676–0.925)
Deep neural network	0.850 (0.793–0.906)	0.846 (0.774–0.917)	0.718 (0.626–0.810)	0.708 (0.573–0.844)	0.817 (0.759–0.874)	0.707 (0.544–0.871)
Logistic regression	0.934 (0.900–0.967)	0.785 (0.642–0.929)	0.929 (0.894–0.964)	0.734 (0.537–0.932)	0.879 (0.823–0.934)	0.827 (0.694–0.960)

^δ^ Removed biomarkers: IL-8, IL-6, angiopoietin-2, E-selectin and VCAM1. ^δδ^ Top five features: total SOFA score; IL-8; D-dimer; platelet and albumin.

## Data Availability

Upon publication of this article, the full dataset and codes will be freely available online in Github, a secure online repository for research data (https://github.com/wujinja-cgu/Sepsis-death-prediction accessed on 24 May 2021).

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
