# Peer review of "Using Machine Learning to Develop and Validate an In-Hospital Mortality Prediction Model for Patients with Suspected Sepsis"

_biomedicines, 2022, doi:10.3390/biomedicines10040802_

Round 1

Reviewer 1 Report

The study is well done and well presented. My questions would be:

  1. Who are the target audience for this paper? In other words is this meant to be a tool for monitoring sepsis outcomes for institutions, communities, regions, countries, etc. at baseline and over time to see if mortality outcomes are changing (improving or worsening)? or do you see this as a tool of interest to clinicians and if so which physicians: emergency, ICU, generalists?
  2. Moving forward it is totally inappropriate to complete this broad array of investigations on all patients with Sepsis 1 and not even all patients in Sepsis 3 or Septic Shock. As such I think you need to take the higher predictive markers to include as a potential package for future researchers to focus on. Your paper seems to outline the following as the keys: SOFA score, IL-8, albumin, D-dimer, Platelets, and AaDO2 +/- lactate (which is in common use) +/- cortisol (also broadly available). SOFA -respiratory is already counted in the SOFA score and also reflected in AaDO2 as are other ABG results. The one thing that is not mentioned as a separate factor, although is part of SOFA is altered mental status and in your study all patients who presented with a GCS less than 15 were in the mortality group. The other key features that you point out as differences among your survivor and non survivor groups are age and malignancy. Do these need to be included in your ML models? Importantly while your model may do well in predicting outcomes at a group level, how would it do at the level of an individual patient, given that many patients within the higher risk groups did survive. Given that, it would seem a bad strategy to patient care to have the clinician be swayed into believing that mortality was likely in a given patient in whom the decision was made to provide aggressive life saving treatment. Reviewing after the fact and noting that someone who died, was in fact in the higher mortality group, is certainly worth discussing at M & M rounds; and may even be worth knowing at a point in time where the decision to continue with aggressive care or to change to a palliative approach may also be a time to bring this into consideration, however, before that point identifying the patient as sepsis or septic shock needing/requesting heroic care is what is needed.
  3. In table S2 you list 20 different biomarkers; in S3 you list 13 different novel biomarkers; and in Table 3 you start with 30 selected features and then remove IL-8, IL-6, angiopoetin-2, E-selectin, and VCAM1. Looking at the 30 features listed in table S4 it seems that the list double counts certain features such as SOFA plus SOFA score-respiratory (you include FiO2 and AaDo2 as separate measures) and SOFA score-coagulation (as platelets is a separate measure), and septic shock and SOFA score cardiovascular, but don't include SOFA score-GCS. Additionally pH, pCO2, Total CO2, SBE, and ABE are interconnected measures as are AaDO2 and FiO2. When you moved from 30 to 25 features my question would be, rather than removing all 5 biomarkers, why not retain IL-8 as it was the most predictive of the unique biomarkers and why don't you remove some of the redundant or double count markers I outlined above to see how that impacts predictability.
  4. Note that in table S4 E-selectin is misspelled !
  5. Given the long list of novel biomarkers included in your study, if only IL-8 were used as the single novel biomarkers to supplement the standard SIRS, qSOFA, SOFA data plus standard labs: CBC, creatinine, electrolytes, glucose, calcium, LFTs, ABGs, albumin, D-dimer, Cortisol, lactate, FDP what would be the predictive value of that? I guess i could be persuaded to include procalcitonin, however, not sure its value but would not include uric acid as part of the standard package, unless good research showing its value.
  6. Your use of Random Forest, RANGER, neural networks, logistic regression, etc. and the predictive differences are beyond my level of understanding, however, for logistic regression (When using a multiple regression model, the predictor variables should be independent and unrelated with one another, but related to the outcome variable.). Within your data set of features, as outlined above, you have a number of variables that are very much related to each other. I would be interested to know the impact of that on the numerous models described in Table 3.
  7. I believe you need to address some of these concerns in the discussion and also outline how you believe this predictive modeling its targeted for and how it should be used. As a reviewer I don't see it being useful to Emergency Physicians, but perhaps for Sepsis 3 patients, could be used as a starting point to derive an ED order set that would benefit ongoing in hospital/ICU care, obviously in conjunction/discussion with Intensivists for their agreement and input. 
  8. If you address some of these questions, then I think you might want to reflect on your responses in the conclusion as all you have said is that using a large number of data points (more than would be reasonable in an efficient, cost effective health care system) your ML model does an excellent job predicting mortality and that future studies are needed to externally validate. The bigger questions are: why? for whom? what features need to be included to be effective and cost appropriate? and why should we care?

Reviewer 2 Report

Good work. I just have two questions:

  1. What is the architecture of the best RF model (how many trees, etc.)?
  2. What software did you use i.e. which packages to work with ML tools -was it Python or R or maybe something else?
